# Measuring the Intermediate Goods' External Dependency on the Global Value Chain: A Case Study of China

## Wen Chen and Lizhi Xing *

College of Economic and Management, Beijing University of Technology, Beijing 100124, China; chenwen@emails.bjut.edu.cn
* Correspondence: itwasa@163.com

**Abstract:** In the face of the anti-globalization trend and the shrinking of the global value chain, ensuring the safety of the global layout of the industrial chain and the sustainability of each country's internal intermediate product production cycle has become an important new development strategy for all countries. The sustainability of the internal and external cycles of production systems is closely related to global value chains. Based on the world input-output model, we define the trade pattern of intermediate goods in various countries from the perspective of trade intermediary attributes, and propose two indicators by which to measure the dependence of China on the global value chain in the process of "dual circulation" development: the degree of vertical specialization (VSD) and the import share of domestic total consumption (IMS); China's super-large market leads to low values of both VSD and IMS. China's high-tech industry has the highest degree of external dependence in the process of participating in dual circulation, and there has been a fluctuation cycle since 2009. The external dependence of different industries shows heterogeneity.

**Keywords:** global value chain; dual circulation; multi-regional input-output table; mediation attribute; external dependency; intermediate products trade

## 1. Introduction

In the context of economic globalization, multinational enterprises integrate production resources that are scattered around the world, according to the resource endowments and comparative advantages of different countries or regions. By reconfiguring each stage in the production process in different countries or regions, an international vertical specialization division of labor system is ultimately formed [1]. Under the existing international division of labor system, the development of a country's or region's production system not only depends on its own final demand and production technology but also on its position in the international division of labor system [2,3]. In this era of globalization, the economies of various countries are interconnected through trade and exchange rates. Therefore, an economic or financial crisis in a major economy can also adversely affect other countries. For example, the occurrence of the subprime mortgage crisis in the United States not only affected the US economy but also caused the market demand in European countries to remain sluggish [4]. Therefore, while actively participating in the global industrial division of labor, countries should also pay attention to the degree of dependence of the industry on the global value chain (GVC).

Global value chains are now undergoing severe shocks. The development of digital technology has reduced labor costs and promoted the return of labor-intensive industries, from developing countries to developed countries. In addition, the world has fallen into the trend of "de-globalization", and the sluggish global value chain has made the environment for countries to participate in the international division of labor even worse. As a result, from the perspective of production, the division of labor in the global value chain, led and promoted by multinational corporations, has shrunk to a certain extent [5,6]. In particular,

the COVID-19 epidemic has highlighted the urgency and importance of maintaining the stability of the supply chain under the impact of "black swan" events [7–9]. In the production of intermediate products, industries that are highly dependent on the global supply chain system will seriously affect economic development once they face a decline in the inventory of intermediate products and a lack of alternative channels. Agriculture and food exports from Australia and New Zealand have been largely caught up in delays to freight services, due to the emergence of COVID-19. Achieving resilience in their global value chains is therefore now a key strategy for companies in Australia and New Zealand. The key drivers of resilient supply chains are domestic production systems and consumer markets, and industrial exporters should consider reducing their excessive dependence on other countries [10].

More and more countries are realizing the importance of industrial chain security. In order to coordinate the safety of the global layout of the industrial chain and the sustainability of the production cycle of intermediate products within a particular country, the country tends to complete a three-stage production cycle domestically. In other words, the cycle of production is intended to increase the domestic production cycle of intermediate products. However, whether or not the transfer of intermediate production to the country can be successful depends to a large extent on the integrity of the current domestic production system and the degree of external dependence. Therefore, analyzing the dependence of the country on the global industrial chain to complete domestic intermediate product production under the existing industrial layout will help to avoid the risks of global intermediate product trade caused by uncertainty, technology blockades, rule decoupling, and other major hidden dangers.

Some recent studies have shown that another important factor driving the restructuring of global value chains is the continued expansion of domestic demand in emerging economies [11]. As the largest representative of developing countries in terms of global trade, China has achieved great success in the past by relying on the open development model of the traditional international cycle. China has not only accumulated enormous wealth from the benefits of the division of labor and trade but has also contributed to the rapid development of productive forces [12]. However, many experts and scholars have pointed out that China faces serious trade friction amid the restructuring of global value chains. In addition, the passive integration of China's manufacturing industry into the global value chain needs to be improved urgently. The agglomeration of domestic industrial chains, huge market advantages, and the deep integration of global value chains are huge advantages for China when developing the internal circulation of the production system [13,14]. Against this background, China has proposed a "dual circulation" policy to deal with tactical adjustments to the internal and external environment. Under the guidance of China's new strategy, the shift om the economic center of gravity and the changes of internal and external driving forces are inevitable. For such a hyper-scale country, changes in domestic production chains have become one of the driving forces behind the restructuring of global value chains. Therefore, studying the degree to which China's intermediate product production system participates in international circulation and domestic circulation gives the degree of external dependence and has become an important subject.

Based on the above viewpoints, we believe that it is necessary to re-examine the dependence of each country's production system on its domestic value chain and foreign value chain, along with its economic significance from a global perspective. Therefore, two scientific problems are to be solved in this paper: the first is to construct a GIVCN network model reflecting the topological characteristics of the global value chain, based on complex network theory and multi-regional input-output (MRIO) data, to restore the industrial sector level of the world's major economies. The second is to measure the degree of vertical specialization of the industrial sector (VSD) and the import share of domestic total consumption (IMS), based on the concept of trade intermediary attributes. The two indicators proposed in this study enrich the theoretical framework of the research on the

sustainability of industrial development. This paper takes the "dual circulation" perspective as its starting point and measures the degree of dependence on China's manufacturing industry using the global value-chain division of labor system. Finally, this paper puts forward relevant suggestions on achieving a two-way balance between foreign countries and Chinese regions and ensuring sustainable development.

## 2. Review of the Literature

### 2.1. Cross-Border Production on GVC

The rise of global value chains has naturally caught the attention of international trade economists, who have worked extensively with ingenious empirical methods to analyze the value-added state of international trade flows and the production system of intermediate goods. In related research, input-output (IO) tables are feasible for measuring standard trade and vertical trade. With the availability and utilization of the global IO database, particularly the inter-country input-output (ICIO) table, quantitative indicators can be constructed to assess its impact on GVC, as it better describes international origins and usage. Scholars have focused on intermediate cross-border production processes in a large number of studies, developing unique methods to measure the function and status of certain sectors in terms of globalization.

Focusing on the use of imported inputs in the production of export goods, Hummels et al. propose vertical specialization—the first empirical measure of participation in vertically specialized trade [15]. Antras et al. derived two different approaches to measure the industrial upstream flow and demonstrated its significant impact on trade flows [16]. Fally et al. quantified the average length of a production chain, reflecting the number of stages required for production and the number of stages between production and final consumption [17]. He and Hillberry then extended the empirical study from cross-factory to cross-border by adopting the IDE-JETRO four-dimensional IO tables [18]. Johnson and Noguera combined IO and bilateral trade data to quantify cross-border production linkages and calculate the value added by bilateral trade [19]. Koopman et al. adjusted all previous measures of vertical specialization and value-added trade to analyze the back-and-forth trade of cross-border intermediates. At the same time, he proposed the global value chain status and participation index to measure the degree of participation of a country's sector in the global production chain. In order to empirically decompose total exports, they constructed a global ICIO based mainly on the seventh edition of the GTAP database [20]. Wang et al. use a new framework to break down total production activities according to whether the relevant value chain is for pure domestic demand, traditional international trade, or simple and complex global value-chain activities [21]. As the ICIO database is updated, we can apply these research frameworks to generate time series that decompose total trade flows into their value-added components.

### 2.2. Measurement of External Dependency

In traditional research, the foreign trade dependence proposed by Grassman [22] is widely used to measure the dependence of a country's economy on other countries. Foreign trade dependence is the ratio of the total import and export volume of a country or region to its GDP at a certain period of time. This indicator has been widely discussed and applied by scholars [23], but some scholars have questioned the fact that the degree of dependence on foreign trade cannot comprehensively and objectively reflect the degree of dependence of a particular country's economy on the international economy [24,25]. In particular, this indicator does not take into account the trade of intermediate goods that gradually occupies a larger share in international trade. Statistics are carried out on the import and export of the same processed goods, thus overestimating the degree of a country's foreign dependence. Research on the accounting of trade value added in global value chains can help to make up for the above shortcomings. To accurately account for the value-added status implied by trade, world input-output models are constructed. This model establishes the relationship between the total output and the final demand of each

country through the endogenous trade of intermediate goods, thereby helping to measure the added value in trade [26]. Daudin et al. [27] proposed the concept of "value-added trade" for the first time and calculated the distribution of added value in the exports of various industries in various countries. After that, the relevant input-output database was also established [28,29]. Based on the measurement of value-added trade, in two separate studies, Bart Los and his colleagues [30,31] used the world input-output model to measure the degree of fragmentation of international trade, based on the measurement of domestic and foreign value-added systems and assessed the changes in European competitiveness. Steen-Olsen et al. [32] compared various world input-output databases by measuring value-added systems in trade and consumption. Patel et al. [33] constructed an effective exchange rate measurement method to evaluate competitive advantage on the basis of the world input-output model, to measure the added value. According to the above research, it can be seen that the use of the world input-output model can avoid the double-counting caused by the trade of intermediate goods, so as to measure the external dependence of trade more accurately. Belke and Wang [34] used the value-added trade volume (the sum of value-added exports and value-added imports) to replace the trade volume of customs statistics, to study the influencing factors of foreign dependence. Larudee [35] used this indicator to re-measure Mexico's foreign dependence and found that the traditional model of foreign trade dependence would overestimate Mexico's foreign dependence.

The existing literature on the division of labor in the global value chain constructs indicators from the perspective of trade added value, which reflects the amount of domestic value contained in export products. The method proposed in this paper is based on social network analysis. Assessing intermediary attributes enables an in-depth study of the cross-border production model in the trade of intermediate goods, which complements the analytical angle based on value-added criteria. Although the above research studies globalization and analyzes the external dependence of value-added trade from a multi-dimensional perspective, there are few studies to quantify and analyze the trade pattern of the internal and external circular production of national intermediate goods and the degree of dependence of different industries on the global value chain. In the case of failing to accurately grasp the basic laws of directional selection inside and outside the economic cycle, local governments are easily caught in the dilemma of a blind layout. This paper can also provide a solid theoretical basis for a particular country to rearrange its industrial chain.

Compared with the existing literature, the innovation presented by this paper is mainly reflected in the following aspects: (1) this study uses a multi-regional input-output model to provide a new measurement framework for the classification of trade patterns, based on intermediary attributes. This measurement framework enhances the study of international trade from an economic physics perspective. (2) Based on the characteristics of the global input-output model, this paper defines the degree of vertical specialization (VSD), the import share of domestic total consumption (IMS) indicators. (3) In response to the "dual circulation" policy proposed by China, the external dependence measurement of the global intermediate product production cycle is extended and the time evolution law of the domestic market, participating in internal and external circulation, is explored. The technical framework provided in this paper is conducive to accurately capturing the key industries under the "dual circulation" strategy. It also puts forward relevant suggestions on reducing the losses caused by the shift of the industrial chain by analyzing the characteristics of the industries in question. The arrangement of this paper is as follows: Section 3 discusses the modeling process of the global industrial value-chain network, Section 4 defines the intermediary attributes of the industrial sector and offers a method for measuring dependence, Section 5 conducts empirical research and discusses our findings, and Section 6 presents the promotion of China's policy recommendations for the implementation of the "dual circulation" strategy.

## 3. Data and Model

In order to represent the nested structure formed by industrial sectors being both providers and consumers of intermediate goods on the GVC, it is necessary to build a GVC network that can reflect the co-opetition relationships of the industrial sectors in each country.

### 3.1. Data Source

In order to facilitate the study of product trade relations among global economies, some institutions have established interregional (international) input-output databases (MRIO or ICIO databases for short). At present, some databases have difficulties in studying Asian economies. For example, the Eora multi-regional input-output database (Eora-MRIO) covers 189 economies but the data provided by it is highly aggregated, so it is difficult to directly use this database to study the topological structure of GVC. In contrast, the world input-output database (WIOD) and OECD-WTO database, based on trade-added value (TiVA) can provide more fine-grained industrial sector information, which can be used widely in empirical analyses focusing on trade. However, the latest data from these databases is only up to 2015 and does not cover all Asian economies.

On this basis, in order to pay more attention to Asian economies, according to the data of the supply and demand table and the input-output table compiled by the Asian Development Bank (ADB) and 18 economies in the Asia-Pacific region from 2008 to 2018, the Asian Development Bank expanded the GVC statistics of Asian economies, based on the WIOD database, thus forming the ADB-MRIO database. This database can provide the latest input-output data, which is helpful for providing relevant statistical data for more Asian economies, to analyze GVC and become an information source for research and policy-making. The ADB-MRIO covers 62 countries and other parts of the world (RoW for short) and 35 industrial sectors, covering the period from 2006 to 2018, as shown in Table 1.

### 3.2. Network Modeling

In this paper, the industrial sector of each country or region is regarded as a vertex, and the input-output relationship between them is regarded as an edge. The value stream reflecting the strength of the relationship is known as the weight. The number of nodes is N, the set of nodes is V, the set of edges is E, and the weight set is W. Thus, the graph G = (V, E, W) is obtained.

Accordingly, this paper establishes a global industrial value chain network model (referred to as the GIVCN model, while the model based on the ADB-MRIO database is labeled as the GIVCN-ADBMRIO model) that depicts the industrial division of labor undertaken by industrial departments in various countries or regions, with the background of global economic integration, and reflects the trade process of various intermediate products in the world in terms of the vertical specialized division of labor. The GIVCN-ADBMRIO model includes 2205 nodes, representing 2205 industrial sectors, of which 2170 nodes belong to 62 countries or regions, respectively, and 35 nodes represent the sum of 35 similar industrial sectors of RoW. In order to facilitate the analysis of the functions and positions of economies and their industrial sectors in GVC, according to the classification standard of ERDI, 35 industrial sectors of the ADB-MRIO are grouped into 13 categories; that is, primary, low-tech, high- and medium-tech, utilities, construction, trade and repair services, tourism, transport services, ICT services, finance and insurance services, property services, public and welfare services, and those services provided by private households. Figure 1 shows the GIVCN-ADBMRIO model based on the aggregated ADB-MRIO data of 2016, 2017, 2018, and 2019. Because the edges of the original network are very dense, it is difficult to observe the key links between industrial departments, so the weak industrial links are deleted, according to a modified Floyd algorithm [36]. In the figure, node colors represent different continents, node sizes represent betweenness centrality, and trade flows are represented as weighted links. Except for Row, SC3 in China and Germany showed greater centrality. As shown in the Figure 1, the thickest links appear between different

industrial sectors in the United States and China, indicating that they generate larger trade flows.

**Table 1.** Classification of industrial departments in ADB-MRIO database.

| Numbering | ADB-MRIO Database Department Category | ERDI Classification | Numbering | ADB-MRIO Database Department Category | ERDI Classification |
|---|---|---|---|---|---|
| S01 | Agriculture, forestry, animal husbandry, and fishing | Primary | S19 | Sales, maintenance, and repair of automobiles and motorcycles | Business services |
| S02 | Mining industry | Primary | S20 | Wholesale and trade (excluding automobiles and motorcycles) | Business services |
| S03 | Food, beverages, and tobacco | Low-tech | S21 | Retail (excluding automobiles and motorcycles); household supplies repair | Business services |
| S04 | Textile raw materials and textile products | Low-tech | S22 | Hotel and catering industry | Business services |
| S05 | Leather, leather products and footwear | Low-tech | S23 | Inland transport | Business services |
| S06 | Wood, wood products and cork | Low-tech | S24 | Waterway transport | Business services |
| S07 | Pulp, paper, printing and publishing | Low-tech | S25 | Air transport | Business services |
| S08 | Coking, refining petroleum and nuclear fuel | High- and medium-tech | S26 | Other support and auxiliary transportation activities; activities of travel agencies | Business services |
| S09 | Chemicals and chemical products | High- and medium-tech | S27 | Post and telecommunications | Business services |
| S10 | Rubber and plastic products | Low-tech | S28 | financial intermediaries | Business services |
| S11 | Other nonmetallic minerals | High- and medium-tech | S29 | Real estate | Business services |
| S12 | Basic metals and metal products | High- and medium-tech | S30 | Machinery and equipment leasing; other business activities | Business services |
| S13 | Mechanical products | High- and medium-tech | S31 | Public administration and national defense; compulsory social security | Public and welfare services |
| S14 | Electrical and optical equipment | High- and medium-tech | S32 | Education | Public and welfare services |
| S15 | Transportation equipment | High- and medium-tech | S33 | Medical and social work | Public and welfare services |
| S16 | Other manufacturing and recycling industries | Low-tech | S34 | Other community, social and personal services | Public and welfare services |
| S17 | Electricity, natural gas and water production | Low-tech | S35 | Household employment service | Public and welfare services |
| S18 | Construction industry | Low-tech | | | |

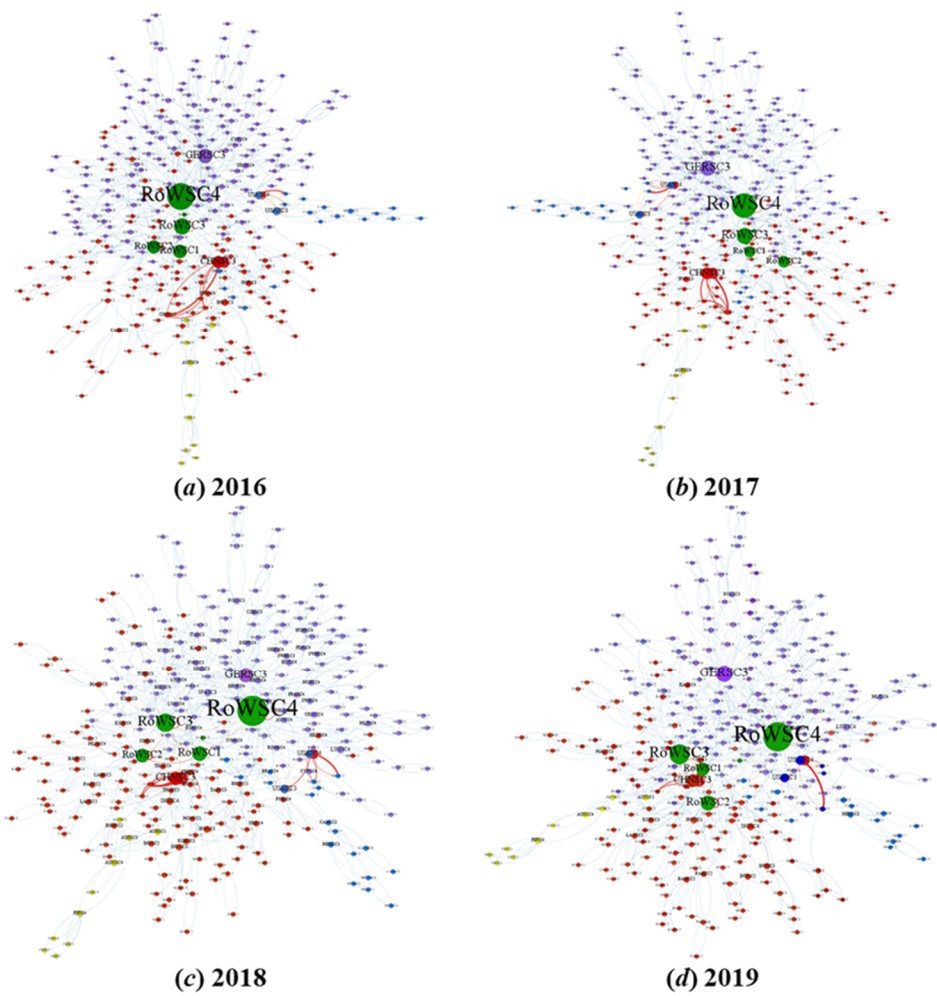

**Figure 1.** GIVCN-ADBMRIO model (aggregated into 5 departments).

## 4. Definition and Measures of Brokerage

In social network analysis (SNA), Marsden defines brokerage as the process in which an actor in a middle position coordinates information transmission [37]. That is to say, in a basic information transmission structure composed of three actors, the actor in the middle is in the intermediary position, so that is the broker. Many empirical studies and theoretical studies mainly regard the intermediary attribute as an important theoretical concept [38,39], but few attempts have been made to quantify it until Gould and Fernandez put forward the formal definition of an intermediary in a social system [40]. They found that if the network can be divided into relatively mutually exclusive subgroups, and if the nodes belonging to different non-overlapping subgroups assume the role of information intermediaries, then these nodes can have five different intermediary attributes or roles, and the information exchange relationship between the two nodes is likely to be heterogeneous. Figure 2 shows five different types of intermediary attributes, among which B plays the role of intermediary, and countries in the same region are represented by the same color.

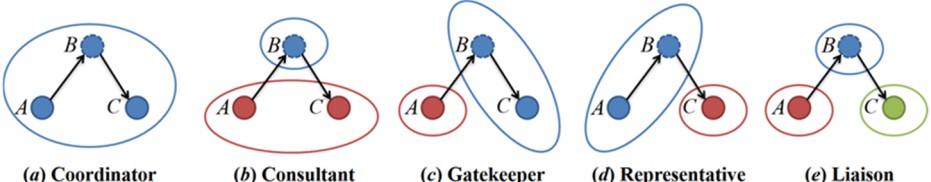

**Figure 2.** Five types of brokerage in the SNA.

If all three players are members of the same group, as shown in Figure 2a, the brokerage relationship is entirely internal to the group, with broker B denoted by "coordinator". If A and C are members of the same subgroup but B is a member of a different group, B is designated as a "consultant", as in Figure 2b. If B belongs to the same group as either A or C, but A and C belong to separate groups, i.e., B brings information into Figure 2c and spreads it out in Figure 2d, the broker is referred to as a "gatekeeper" or "representative", depending on its function. A, B, and C are the roles denoted by the "liaison" in Figure 2e, respectively [41].

### 4.1. Trade Types on the GVC

While international trade has increased substantially over the previous half-century, the structure of trade has changed dramatically as well. One of the most significant changes is the growing interconnectivity of industrial processes in a vertical trading chain spanning many countries, each specializing in different stages of production. Vertical specialization is defined as when a country imports intermediate goods as inputs from another country and then transforms those inputs into value-added outputs that are exported to a third country. This production process entails a slew of domestic and international trade in intermediate commodities, and the process does not end until the final product reaches the consumer market [41].

Trade can be divided into four types, based on whether or not commerce and trade economics take place within the same country and whether or not products and services are exchanged between the various sectors. Then, based on two dimensions, as indicated in Figure 3, a detailed classification can be made: inter-industry trade, intra-industry trade, industrial input–output trade, and industrial self-consumption trade [41].

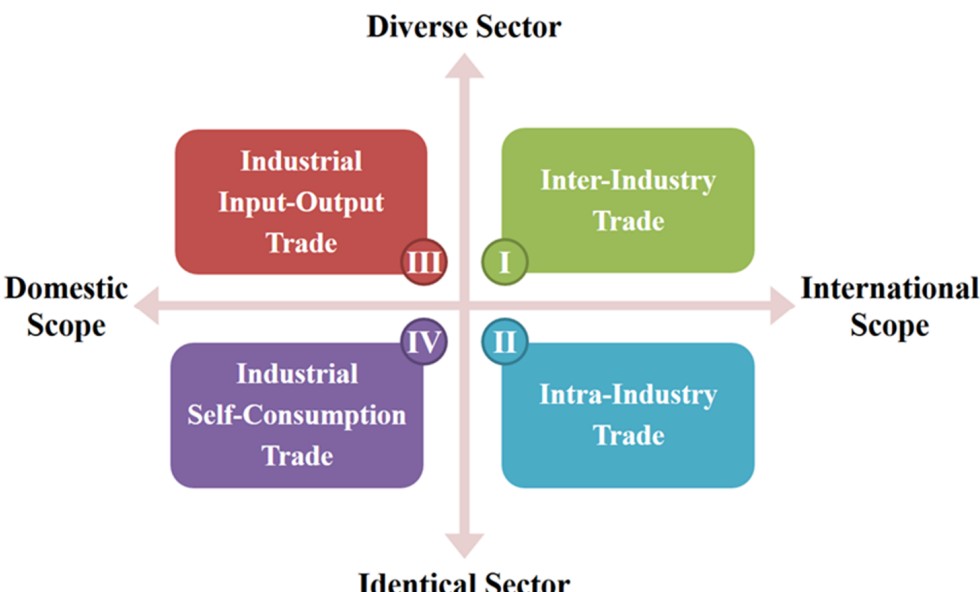

**Figure 3.** Four trade types in their quadrants.

### 4.2. Decomposition of Trade Roles

As revealed by the analysis on brokerage in SNA, five types of trade brokerage property (TBP) in GIVCN-ADB2019 models are proposed: inland trade, international trade I, import trade, export trade, and international trade II, as shown in Figure 4.

In each trade situation, Figure 3 depicts four pairings of factors: intra-country or inter-industry transfer, upstream or downstream sector, import or export goods, and inputs or outputs. In each sub-figure, country B's sector k plays a different function in trade, and a thorough explanation of this sector may be found in this section.

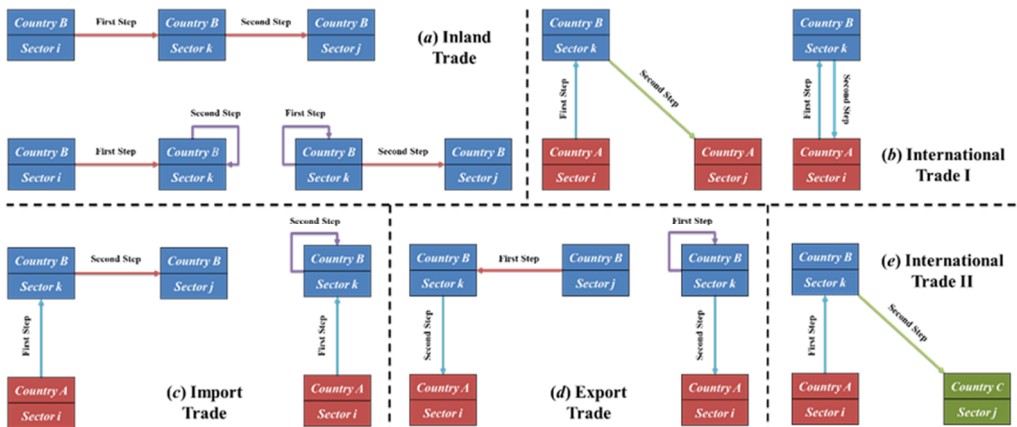

**Figure 4.** Decomposition of trade brokerage property in the GIVCN model.

Inland trade, denoted by *TBP*1, means that sector k and its upstream and downstream sectors all belong to the same country; i.e., both the inputs and outputs of sector k are within the economic system of a single country, as shown in Figure 4a. In addition, self-consumption can be treated as either the inputs from the upstream sector or outputs to the downstream sector.

International trade I, denoted by *TBP*2, means that sector k's upstream and downstream sectors belong to another country; i.e., sector k imports its inputs from another economic system and then exports their outputs back to it (just like OEM), as shown in Figure 4b. It does not matter whether the overseas providers and consumers are the same. Besides this arrangement, *TBP*2 also includes a certain proportion of re-import and re-export trade.

The import trade, denoted by *TBP*3, means that sector k and its downstream sector belong to the same country, or to itself as the downstream sector consumes part of its output, while its upstream sector belongs to another country; i.e., the inputs are imported from the overseas market and are then sold inside the home market after the value-added process.

The export trade, denoted by *TBP*4, means that sector k and its upstream sector belong to the same country, or itself as an upstream sector providing part of its input, while its downstream sector belongs to another country; i.e., the inputs are acquired inside the home market and are then exported to the overseas market after the value-added process.

International trade II, denoted by *TBP*5, means that none of sector k or its upstream and downstream sectors belong to the same country. This model can better reflect the character of vertical specialization than *TBP*2. For example, intermediate goods may be produced in Japan and then shipped to China for assembly into final goods that will be consumed in the United States; the relevant sector in China plays the role of TBP5 in this process.

*4.3. Statistical Inference on TBPs*

*TBPs* are responsible for all processes in the vertical specialization and value-added manufacturing process, and sectors always have multiple *TBPs*. As we know, the ICIO data for modeling the industrial chain network is far too broad and complex to directly quantify the *TBPs* of each sector, acting as brokers in a large number of separate local economic systems (overlapping ego networks). Furthermore, simply examining the *TBPs* of each sector one by one is pointless. The suggested research approach is based on probability sets, which characterize the *TBPs* ratio of sectors that are located on the GVC. The probability distribution of *TBPs* is derived as follows [41].

Step 1: We assume that $m$ countries/regions ($u, v = 1, 2, \dots, m$) in the GIVCN model constitute a complete set, as denoted by $\{R_u\}$.

Step 2: We then assume that all $n$ sectors within one country/region constitute $\{z_k\}$ when they play the role of broker on the GVC, i.e., the sector $k$ denoted by $z_k$ is on the

midstream level of IVCs. In which case, $k \in \tau(u)$ and $\tau(u)$ are a set of numbers standing for the row sequence number of a certain country/region in the adjacent matrix $Z^{uv}$. For instance, China is the 8th nation in ADB2019, and the United States is the 43rd one, so $\tau(8) = \{246, 247, \cdots, 280\}$ and $\tau(43) = \{1471, 1472, \cdots, 1505\}$ since each economy owns $n = 35$ sectors.

Step 3: We then assume that the upstream sectors of $z_k$ (which provide raw materials or intermediate goods to $z_k$) constitute $\{s_1, s_2, \cdots, s_{a_u}\}$ and its downstream sectors (which consume intermediate goods from $z_k$) constitute $\{t_1, t_2, \cdots, t_{b_u}\}$. In which, $a_u = max(\tau(u))$, and $b_u = max(\tau(u))$ too.

Thus, the relationship of any sector belonging to the upstream set on the IVCs could be presented as:

$$\{s_1, s_2, \cdots, s_{a_1}\} \subseteq R_1,$$
$$\cdots\cdots$$
$$\{s_{a_{u-1}+1}, s_{a_{u-1}+2}, \cdots, s_{a_u}\} \subseteq R_u,$$
$$\cdots\cdots$$
$$\{s_{a_{m-1}+1}, s_{a_{m-1}+2}, \cdots, s_{a_m}\} \subseteq R_m.$$

Similarly, the relation of any sector belonging to the downstream set on the IVCs could be presented as:

$$\{t_1, t_2, \cdots, t_{b_1}\} \subseteq R_1,$$
$$\cdots\cdots$$
$$\{t_{b_{u-1}+1}, t_{b_{u-1}+2}, \cdots, t_{u_1}\} \subseteq R_u,$$
$$\cdots\cdots$$
$$\{t_{b_{m-1}+1}, t_{b_{m-1}+2}, \cdots, t_{b_m}\} \subseteq R_m.$$

Step 4: Given that $\forall \{z_k\} \subseteq R_u$, $A_u$ denotes sector $k$ in the upstream set belonging to $R_u$ to some degree, while $B_u$ denotes sector $k$ in the downstream set belonging to $R_u$ to some degree. Then, this kind of affiliation relation would be quantified by probabilities of events $A_u$ and $B_u$, as follows:

$$P(A_u) = \frac{\sum_{i=a_{u-1}+1}^{a_u} w_{s_i z_k}}{\sum_{k=1}^{a_m} w_{s_k z_k}} = \frac{\sum_{i \in \tau(u)} w_{s_i z_k}}{\sum_{k=1}^{N} w_{s_k z_k}} \tag{1}$$

where $w_{s_i z_k}$ is the weight of the edge, starting from $s_i$ and reaching to $z_k$. $\sum_{j \in \tau(u)} w_{s_i z_k}$ represents the gross value of the intermediate goods from all the upstream sectors to the midstream sector $k$ when they are in the same country/region $R_u$, and the function of the denominator is to normalize the formula:

$$P(B_u) = \frac{\sum_{j=b_{u-1}+1}^{b_u} w_{z_k t_j}}{\sum_{k=1}^{b_m} w_{z_k t_k}} = \frac{\sum_{j \in \tau(u)} w_{z_k t_j}}{\sum_{k=1}^{N} w_{z_k t_k}} \tag{2}$$

where $w_{z_k t_j}$ is the weight of the edge, starting from $z_k$ and reaching to $t_j$. $\sum_{j \in \tau(u)} w_{z_k t_j}$ represents the gross value of intermediate goods from the midstream sector $k$ to all the downstream sectors when they are in the same country/region $R_u$, and the function of the denominator is to normalize the formula.

Step 5: Obviously, events $A_u$ and $B_u$ are mutually independent. Considering the definition of brokerage and the above conditions, the probability distribution of each *TBP* is:

$$P\left(z_k^{TBP1}\right) = P(A_u \cap B_u) = P(A_u)P(B_u) \tag{3}$$

$$P\left(z_k^{TBP2}\right) = \sum_{u=1,v \neq u}^{m} P(A_v \cap B_v) \tag{4}$$

$$P\left(z_k^{TBP3}\right) = [1 - P(A_u)]P(B_u) \tag{5}$$

$$P\left(z_k^{TBP4}\right) = P(A_u)[1 - P(B_v)] \tag{6}$$

$$P\left(z_k^{TBP5}\right) = \sum_{v=1, v\neq u}^{m} P(A_v)[1 - P(B_u) - P(B_v)] \tag{7}$$

where five sorts of probability delegate the ratio of roles that that certain sector plays on the GVC, in detail, $P(z_k^{TBP1})$ stands for inland trade, $P(z_k^{TBP2})$ for international trade I, $P(z_k^{TBP3})$ for import trade, $P(z_k^{TBP4})$ for export trade, and $P(z_k^{TBP5})$ for international trade II.

### 4.4. Measurement of Dependency

The notion of GVC aids in the expansion of ICIO data applications and allows for the measurement of interdependence between the different industries in different nations. The majority of the related research focuses on two aspects. On the one hand, some of them calculated the direct import share of production or total investment, to account for the foreign intermediate goods employed in domestic production. The $S_M$ index, for example, was proposed by Feenstra and Hanson to measure the degree of manufacturing outsourcing [42]. Scholars, on the other hand, have emphasized the value of imported inputs incorporated in exported goods, such as the VS index proposed by Hummels in 2001, which applies to those instances where production is finished in two or more nations and commodities cross the border at least twice [43]. We will attempt to redefine these two sorts of indices from the standpoint of SNA in this section.

*TBP*1 and *TBP*3 both indicate the importance of an industrial sector in promoting domestic downstream output. The distinction, however, is whether the obtained intermediate goods are from inside or outside the country, which represents the extent to which it relies on local and foreign industrial chains while performing value transformation. We design a new model based on *TBP*1 and *TBP*3 to quantify the **import share of domestic total consumption**, denoted as IMS, using lessons learned from the $S_M$ expression form. It is established using the following formula:

$$IMS(k) = \frac{P\left(z_k^{TBP3}\right)}{P\left(z_k^{TBP1}\right) + P\left(z_k^{TBP3}\right)} \times 100\% \tag{8}$$

where *IMS(k)* reflects the import share of a specific sector's overall consumption while providing intermediate products for its domestic IVC.

*TBP*2 and *TBP*5 both illustrate the industrial sector's outsourcing role in the worldwide multi-stage manufacturing process. Although there is a variation in where the upstream and downstream sectors are located, whether in the same country or in different countries, the middle sector is the same. As a result, we combine *TBP*2 and *TBP*5 to calculate VSD, or the **vertical specialization degree**. It uses the following formula:

$$VSD(k) = P\left(z_k^{TBP2}\right) + P\left(z_k^{TBP5}\right) \tag{9}$$

where *VSD(k)* reflects the frequency with which a specific sector $k$ is in the middle of several three-stage IVCs, its imported intermediate goods being used to manufacture the export ones.

In conclusion, from two perspectives, *IMS* and *VSD* represent how much a specific sector/economy relies on foreign trade when it participates in the vertical specialization. The first shows its reliance on imported intermediate goods (import trade), and the second is its reliance on the output of domestically produced intermediate products (export trade).

## 5. Results and Discussion

According to the input-output model established above and the dependence measurement index, this paper calculates the external dependence of a country and its industry. First, it analyzes the situation of many countries participating in the economic cycle at home and abroad, and then explores the specific situation of each industry in China and the reasons behind the phenomenon.

### 5.1. Measurement of Double-Cycle External Dependence

By aggregating the industrial sectors, we focus on calculating the VSD and IMS for primary (P) sectors, low-tech (LT) sectors, and high- and medium-tech (HMT) sectors for 63 economies. The distribution of VSD and IMS across countries/regions is observed by dividing the scatter plot into four quadrants. The results are shown in Figure 5.

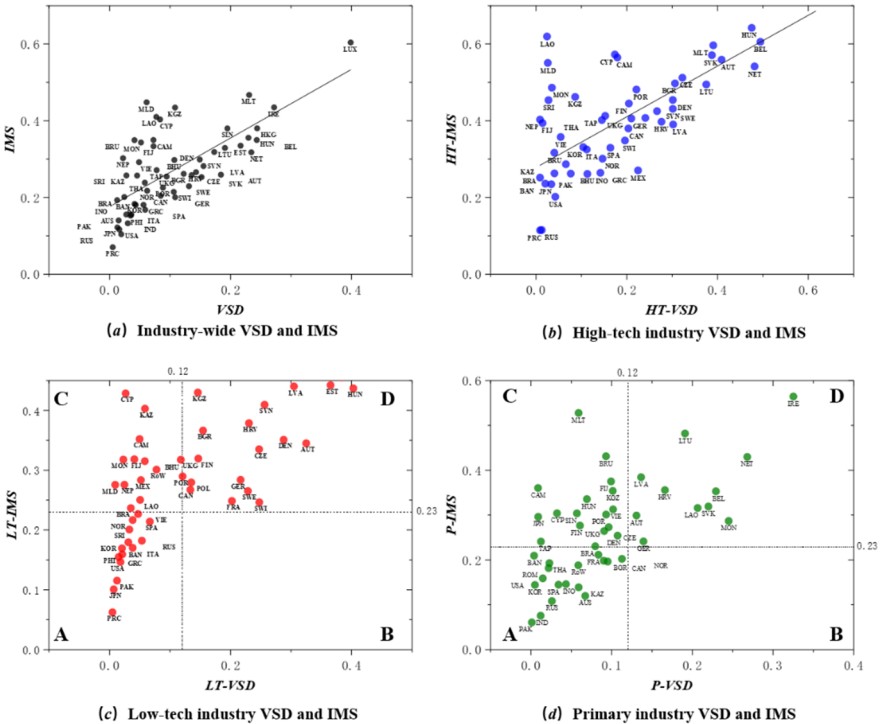

**Figure 5.** IMS and VSD in the industries sector.

The overall VSD and IMS distribution of all sectors in 62 countries and regions globally is shown in Figure 5a. After fitting the scatter plot of industry-wide VSD and IMS, it was found that industry-wide VSD and IMS are positively correlated, which indicates that the external dependence of the external and internal cycles of the production system of most countries behaves consistently. In other words, the more deeply a country participates in the global value-chain division of labor, the more it relies on the import of intermediate goods from abroad to complete the process of intermediate goods production within it. We all know that in today's globalization of trade, it is almost impossible for a country to be independent of global value chains, and no country can have a complete industrial chain to satisfy all domestic demands. Different countries have found their place in global value chains. Less developed countries, for example, tend to appear at the end of GVCs, completing the least value-added part of GVCs by undertaking much of the processing and assembly work. This large concentration of labor-intensive industries inevitably leads to the insufficient mastery of national core technologies, resulting in a deepening of national dependence on GVCs. As a result, numerous studies have emerged on how to achieve value-chain climbing in developing countries. However, when scholars develop their analysis of the degree of participation of countries/regions in GVCs, they ignore the fact that the degree of participation of those countries in completing the internal cycle of intermediate goods production and the global links are closely related [6].

The conclusion that IMS is positively correlated with VSD is most evident in the HMT sector, as shown in Figure 5b, where the fitted line shows that IMS and VSD are positively correlated in the HMT sector in most of the 62 countries worldwide, except for a few countries that deviate from the fitted line. The HMT sector has the longest chain and the most complex production process and, therefore, requires more countries/regions to collaborate in production. We find that the deeper a country's involvement in the

HMT sector in global three-stage production, the stronger its external dependence on the domestic production cycle, a pattern that is consistent with reality. Countries such as the UK, Canada, Germany, and Brazil have been able to maintain a high balance of external dependence on domestic and foreign intermediate production. For example, Germany has always maintained its technological level in the world; it is the largest technology owner and exporter in Europe, and it maintains a high export rate of HMT products while importing a large number of intermediate HMT products from other countries. Of course, not all countries have maintained a balance of foreign dependence, such as the United States, China, and Russia. Both the VSD and IMS of HMT in these countries are relatively low, and the commonality of these countries is that they have a huge domestic market and a complete domestic industrial system. As a result, the share of domestic trade (TBP1) is much larger than that of other countries, thus reducing the dependence on GVC. Japan's VSD and IMS are also not high, and we believe that Japan has a clear advantage in high technology industries and has long implemented local protection policies, so the internal cycle is less dependent on foreign countries. Meanwhile, some countries have a very high dependence on GVC in HMT, while IMS in the HMT sector is much larger than VSD, e.g., in Laos, Cyprus, and Cambodia, where the domestic HMT intermediate production systems are relatively backward, and the domestic production of high-technology products depends largely on foreign imports, limiting their economic development level to a great extent.

In Figure 5c, most countries are distributed within Region C (low VSD–high IMS) and Region D (high VSD–high IMS). This indicates that the LT sector has the highest global liquidity of all the industries. The LT sector has a relatively low technology content and the technology transfer between countries is very common. Countries in region D have a high degree of external dependence on the domestic and foreign production of intermediate goods and, by observation, most of these countries are EU member states. The national economies are small, and their national industrial chains are deeply embedded in the regional value chains in Europe, forming a community of interests. As a result, these countries are closely connected to produce competitive LT products through their comparative advantages, which are thus circulated globally. Take Germany, an EU member state, for example; it is all located in Region D (high VSD-high IMS) in the LT industry distribution map, is highly involved in foreign economic cycles, and has a high degree of foreign dependence on domestic economic cycles, which is also in line with Germany's economic state of deep involvement in the development of global value chains.

The basic industries mainly include agriculture, forestry, and mining, which have short value chains and circulate globally as the most basic products. In Figure 5d, most countries are distributed in Region A (low VSD–low IMS) and Region C (low VSD–high IMS). The lower VSD implies that the global flow of resource products is not high, and most countries are able to complete the entire production process of basic products within their domestic production systems. Some countries with relatively scarce domestic resources need to import large amounts of basic intermediate products to complete the subsequent production stages and, thus, meet the domestic demand for the final product. Some countries rely on the import of resource-based intermediate goods, such as food and minerals (e.g., Singapore, Thailand, etc.), reflecting a high level of IMS, while fewer countries show a high level of dependence in both the internal and external cycles of their economy. Similarly, Japan's distribution of the basic industries in Region C implies a high dependence on foreign countries when basic products participate in the domestic cycle and low participation in the external cycle of basic intermediate goods, which is consistent with the country's economic characteristics. Japan's domestic resources are extremely scarce and, as the second-largest energy consumer after the United States, Japan relies heavily on resource imports and exports fewer resource-intermediate goods.

*5.2. Measurement of China's Double Cycle External Dependence*

This section further analyzes the dependence of China's basic industries, low-tech industries, and high-tech industries on the GVC in domestic circulation and foreign circula-

tion. Figure 5 shows the numerical values and changing trends of the VSD and IMS of the three industrial sectors.

The VSD of China's three sectors is shown in Figure 6a, and the VSD of high-tech industries is higher than the overall level of the other two industries, indicating that China's high-tech industries are highly dependent on external circulation. China's medium- and high-tech industry is mainly based on the processing trade policy of "big in and big out, with two ends out", As early as 2007, the output value of its high-tech products ranked second in the world and it became a veritable "world factory" of high-tech products. However, behind the explosive increase of the import and export of high-tech products, China is faced with the risk of being "dismembered" under the international vertical division of labor [44]. Their overall VSD of basic industries is the lowest because China is a populous country. Compared with high-tech products, the manufacturing of basic industries has a shorter value chain and is more to meet domestic demand [45]. The VSD of the low-tech industry is in the middle position, which mainly includes paper, textile products, leather processing and manufacturing, etc., These industries mainly adopt the general trade mode and the processing trade is significantly reduced. With its domestic raw materials and labor, China has completed the main value-added part of the process, so its participation in external circulation is less than that in high-tech industries [46].

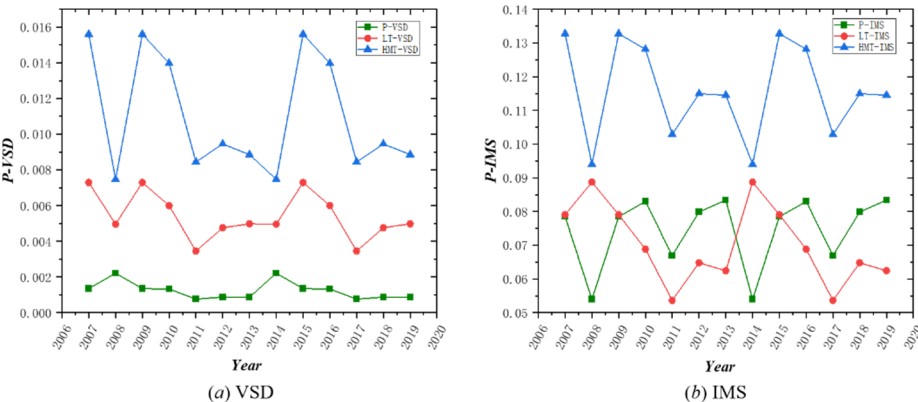

**Figure 6.** The changing trend of VSD and IMS in three types of departments in China. (Blue lines represent high-tech industries, red are low-tech industries, and green are basic industries).

From the perspective of its changing trends, the basic industry changed and only increased slightly in 2008 and 2014. The changing trend of medium- and high-tech industries is basically the same as that of the low-tech industries; it dropped to a low point in 2008 and 2014. China's medium- and high-tech industries are deeply embedded in the global value chain and have a higher dependence on external circulation. Therefore, when the foreign economic environment changes, medium- and high-tech industries will be more impacted, showing a greater range of changes in the figure. After China's accession to the WTO, foreign trade has developed rapidly and the export trade volume has increased from CNY 2063.44 billion in 2000 to CNY 13,713.14 billion in 2013, surpassing the United States for the first time and becoming the country with the largest foreign trade volume in the world [47]. However, in 2008, the US subprime mortgage crisis broke out and swept the world rapidly. The global financial crisis and economic downturn caused the consumer confidence index of developed countries to drop sharply, while the international demand was weak, which caused a great blow to China's foreign trade exports. Therefore, the VSD of low-tech industries and high-tech industries both dropped to their lowest point in 2008. In 2008, China's dependence on foreign countries was greatly reduced passively, which also made the Chinese government realize that the country's high dependence on foreign countries was associated with poor economic stability, so it began to gradually change the economic growth model and expand domestic demand. With the stimulus policies of various countries and its slow economic recovery, China recovered to the level before the financial crisis in 2009–2010, while VSD also increased accordingly [48]. Since 2010, China's

economic growth has slowed down and its structural contradictions have become more prominent, China has further adjusted its industrial structure, and the corresponding VSD is at a low level. The VSM of high- and low-technology industries was at its lowest level in 2014. Since then, China has entered a period of increasing speed, shifting gears, and experiencing structural transformation.

The IMS of three types of departments in China is shown in Figure 6b; the IMS of high-tech products is higher than that of low-tech industries and basic industries, which proves that companies upstream of the industrial chain have the highest dependence on foreign economic circulation when high-tech intermediate products participate in the domestic circulation process. IMS, basic industry, is in the middle position, indicating that in the process of the domestic economic cycle, trade and production activities have a high degree of dependence on external resource-intermediate products. The basic industries are mainly agriculture and mining, among which the importing of intermediate products of mineral resources can greatly alleviate the constraints of China's shortage factors on development. In contrast, low-tech industries have the lowest IMS; that is, they have the lowest dependence on foreign intermediate products in terms of domestic circulation. This is because China has the advantages of industrial clusters with strong supporting capacity and high efficiency in the textile, leather, and other manufacturing industries, and is in an upstream position in the global value chain.

From the perspective of changing trends, the changes in low-tech industries and the basic industries are fundamentally the same; since 2009, there has been a phenomenon where there is a periodic bottoming-out every three years, with a short-term rise in 2009, 2012, and 2015, respectively, and then continued to decline. On the one hand, with the deepening of the vertical division of labor in globalization, the fluctuation of the world economy has an impact on China's foreign trade, while the world economy itself is cyclical; on the other hand, the Chinese government deepens the adjustment of the industrial structure, expands domestic demand, and promotes the domestic economic cycle, The short economic cycle is the result of China's rebalancing of the import and export trade. The IMS fluctuation of low-tech industries is relatively small, with small peaks in 2008 and 2014, which is opposite to the changing trend of basic industries and high-tech industries. In 2008, the global financial crisis passively reduced the dependence of China's economic internal circulation on foreign countries, and the IMS of high-tech industries and basic industries declined accordingly. In order to make up for China's losses in these industries, China exerted its advantages in the technology industries, expanded its domestic market, and increased its imports of upstream intermediate products, which caused the IMS to rise. In the early stage of 2014, due to the influence of the abnormally high base of trade, foreign trade experienced a double decline in import and export that had not happened in many years, At this time, the domestic investment demand slowed down, and the dependence on internal circulation of the country declined, Under these circumstances, low-tech industries, including leather manufacturing, textiles, and garment manufacturing, actively expanded their markets and realized the growth of trade value [49].

*5.3. Discussion*

This paper proposes a new measurement method to measure the external dependence of the three major industries in 62 countries around the world; the main conclusions obtained have also been confirmed in other articles. We found that the primary industries in 62 countries around the world, including agriculture, forestry, animal husbandry, and fishery, have the lowest participation in foreign production cycles, while domestic production links are less dependent on global value chains. In response to this phenomenon, we recognize that the primary industries are due to the low level of opening up to the outside world, the low technical content, and the low degree of integration with other industries. Agriculture and animal husbandry are more involved in the downstream production activities of the global value chain, while the number of simple global value-chain activities is larger than for complex global value-chain activities. For less-developed countries, em-

bedding the basic industries in global value chains is a staple of national agricultural and industrial policies. Gammelgaard et al. analyzed Malawi as a research object and found that the manufacturing of basic industrial products lacked scale effects at the community level, while a large number of products only flowed into the domestic market [50]. This pattern is also confirmed in Acharya's research on Nepal's main agricultural export product. Although the currently developing agricultural industry chain has been linked to the global industry chain, the export of basic products faces strict food safety regulations, high tariffs, and complex Customs clearance procedures [51]; therefore, from an overall perspective, most of the basic industry products are completed in the domestic value-added sector, which is less dependent on the global industrial chain.

Zhou and Yu found in their analysis of China's basic industries that the value-added status of basic product processing in China is not high; it mainly depends on the primary processing of raw resources. Furthermore, it lacks the depth of the deep integration of natural resources with advanced technology [52]. He and Yang believe that, from the emergence of market demand to the satisfaction of that market demand, the basic product industry chain completes a simple cycle in China. Therefore, they proposed that under the "dual circulation" scenario, China should strengthen the smoothness of the cross-regional circulation of the agricultural industry chain [53,54]. In general, the above research is consistent with the conclusions drawn in this paper, regarding an analysis of the dependence of basic industries on global value chains.

Different from previous papers, this paper finds that basic industries and low-tech industries have cyclical fluctuations. The discovery of this phenomenon reveals the linkage response of my country's manufacturing industry to changes in international trade. There are more complex mechanisms behind cyclical fluctuations and the linkage phenomenon in international trade, mechanisms that this paper has not explored in depth, which is also a shortcoming of this paper.

In the analysis of high-tech industries, it was found that China's high-tech industries have the highest participation in domestic and foreign production cycles and are more responsive to changes in the global situation. Because China relies on its cheap labor cost advantage to be locked into the low-end processing and assembly links of the global value chain. With the loss of China's cheap labor cost advantage, China's high-tech industries may face the risk of being "extruded" from the global value chain. This view is consistent with previous research.

With profound changes in the international pattern, the reconstruction of the global value chain is an objective necessity and the high-tech industry is more likely to undergo reconstruction than other industries [55]. The importance of high-tech industry in a national industrial layout and its role as a driving force for economic growth has been studied by many recent scholars [56–58]. Gaulier et al. pointed out that China is not only the location of the final stage of production but is also in the middle stage of the value-added chain, with the final products accounting for less than half of its high-tech exports [59]. Not only that, China's imports of high-data intermediate goods are used for export product processing activities, rather than to meet the final product needs of the domestic high-tech industry [60]. Lemoine et al. believe that the export of most high-tech products also occurs in terms of components [61]. The above studies confirm that both the domestic and foreign production cycles of China's high-tech industries are overly dependent on global value chains. This is not only the case in China; Stöllinger et al. also found this phenomenon when they analyzed the foreign dependence of EU high-tech industry trade [62]. Due to the complex production processes and high R&D cost of high-tech industries, in this study, many countries have shown high external dependence, especially some of the less developed countries.

Compared with the previous research, this study also offers some new findings. The high-tech industry VSD and IMS are positively correlated, which shows that the external dependence on the external and internal production cycles of most countries is consistent. The characteristics of high-tech industries determine the fact that the higher the degree of

a country's participation in foreign economic cycles, the higher its external dependence on the process of domestic production cycles. This is because, in high-tech industries, all parts of the products need to reach a higher level. This requires high-tech integration from around the world. The higher the degree of the vertical division of engineering in the global value chain for a country that participates in high-tech manufacturing, the more likely it is that it will not be able to complete the closed internal cycle easily.

In past research on global value, an emphasis has been placed on the characteristics of countries embedded in the global value chain. The existing research mainly uses the case study method [63], the domestic value-added rate in exports [64–66], the vertical specialization method [26,43], and the value-added trade accounting method [27] to analyze the embedded characteristics of industries in the global value chain. The VSD index proposed in this paper can measure a country's vertical specialization degree in GVC, enriching the original theoretical method system. More recently, due to the shrinking of the global industrial chain, many countries have shown a trend of completing the processing of intermediate products, from global processing layout to domestic processing. So far, no research has been proposed to specifically measure the dependence of the domestic intermediate product production cycle on the global value chain in response to this phenomenon. The IMS index proposed in this paper measures the import share of an industrial sector in terms of total domestic consumption, which helps a particular country to fully grasp the shortcomings of that country's different industries in the production of intermediate products when making an industrial chain transition strategy. One defect of the indicator proposed in this paper is that it does not distinguish between horizontal trade and vertical trade, and more constraints need to be added to the TBPs formula in the follow-up research.

## 6. Conclusions

At present, some scholars attach great importance to the measurement of cross-border production and external dependence. In order to accurately analyze the new characteristics of international production organizations and to collect and interpret international trade statistics, a large number of analytical methods have been designed. However, in the face of the trend of anti-globalization, there is no indicator to measure the degree of dependence of the country's production cycle, either at home or abroad. This paper makes up for the deficiencies in this field. According to the changes in global value-chain activities, a new framework is used to decompose the total production activities and analyze the unique and effective dependency measure that is designed. This paper uses the latest input-output data to establish an input-output model for intermediate products, adopting the concept of the intermediary role in social network analysis to classify trade patterns, following the "dual circulation" strategy proposed by China as the research object to measure foreign dependence. It closely integrates the domestic economic cycle with the global value chain and analyzes it from two levels of country and industry. The conclusions are as follows:

(1) IMS and VSD satisfy a positive correlation between the foreign dependence of high-tech industries in many countries in the world; that is, the higher the degree of a country's participation in foreign economic cycles, the higher its foreign dependence in its domestic economic cycles. In the distribution map of basic industries and low-tech industries, each country's region reflects its own trade characteristics. Compared with other countries, China has a lower overall VSD and IMS. China's rare, super-large domestic market reduces its degree of foreign dependence in the economic cycle, which is an important advantage to China's development of the "dual circulation" strategy. Leading the development of international trade with domestic demand is an inevitable path for the development of a major trading country. China should expand its domestic market demand, improve the scale and quality of its imports, and gradually form an international market that is synchronized with world rules and standards.

(2) From this analysis of the external dependence of various industries, it can be seen that the high-tech industry has the highest external dependence. Once the global demand is rapidly shrinking, such as the 2008 financial crisis, the external dependence of China's industrial chain will be massively reduced. The industry will be affected to the greatest extent, which contains a risk warning that China is insufficient in original innovation and independent innovation. Therefore, for industries that are highly dependent on export-oriented technology, China should focus on breaking through key technology shortcomings and speeding up the cultivation of a number of supporting enterprises with high innovation levels and strong supply capacity. We should focus on cultivating new fields and new formats, such as 5G and artificial intelligence. In the next five years, a manufacturing production chain dominated by China will be formed, along with multiple emerging economic clusters.

(3) With the increase of China's economic size, the relative shortage of natural resources becomes more prominent. The degree of dependence of basic industries in terms of internal circulation and external dependence can reflect such problems. With the acceleration of national urbanization, the rapid development of heavy chemical industry and infrastructure construction, and the demand for energy and mineral products have increased rapidly. Therefore, in the process of the domestic economic cycle, the dependence on the importing of intermediate products of external resources gradually deepens. However, the import of this important strategic resource is too dependent on foreign countries, and its proportion is too large, which is not conducive to the sustainable development of China's economy in the future. The import of mineral products should not blindly rely on imports. The scale and growth rate of imports are configured from the perspective of new energy. China should focus on promoting a reasonable adjustment to the scale and proportion of its imports through energy and resource product price reforms and tariff adjustments.

(4) For low-tech products, China's increasingly sound integrated supporting system and high level of trade competitiveness have supported China to become a major country in the foreign trade industry. Although China's ability to upgrade production technology and resource allocation continues to improve, it is still shackled at the bottom of the value chain. Therefore, China should attach importance to the upgrading of technology-driven product competitiveness and enhance the technology and R&D capabilities of local enterprises. By improving the international competitiveness of China's manufacturing industry, it will climb to the ends of the chain on both sides of the "smile curve".

**Author Contributions:** Conceptualization, L.X.; methodology, L.X. and W.C.; software, L.X.; validation, W.C.; formal analysis, L.X. and W.C.; investigation, L.X.; resources, W.C.; data curation, W.C.; writing—original draft preparation, W.C.; writing—review and editing, L.X. and W.C.; visualization, W.C.; supervision, L.X.; project administration, L.X.; funding acquisition, L.X. All authors have read and agreed to the published version of the manuscript.

**Funding:** This research was funded by the National Natural Science Foundation of China (Grant No. 71971006) and Technology Plan Key Program of Beijing Municipal Education Commission "Research on the Carbon Neutralization Path of the Beijing-Tianjing-Hebei Region Based on the Analytical Framework of Complex Socio-Economic Networks" (Grant No. 22JI0002).

**Institutional Review Board Statement:** Not applicable.

**Informed Consent Statement:** Not applicable.

**Data Availability Statement:** We signed a confidentiality agreement with the transportation company who provided us with the data used in this work. Hence the data will not be shared.

**Conflicts of Interest:** The authors declare no conflict of interest.

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
