# Peer review of "Measuring the Intermediate Goods’ External Dependency on the Global Value Chain: A Case Study of China"

_sustainability, doi:10.3390/su14074360_

Round 1
Reviewer 1 Report
Strengths of this manuscript:
1. Strong methodology and empirical part of the article.
Weaknesses of this manuscript:
- The title of the article is too long and it is very narrow that would suggest the article is not interesting for international readers.
- This presents the China case mainly and broader perspective and separate literature review part is needed.
- The introduction part focuses on politics mainly. This should be avoided to quote political statements in a scientific article.
- The scientific problem and the novelty contribution to the science should be better presented.
- The topic of the manuscript refers closely to some current issues, e.g. Covid-19 pandemic impact on global trade and production. The current Covid -19 pandemic issue should be better described and extended in the manuscript. The Covid-19 issue is mentioned once in the article and this even not refer to the scientific research (no reference is provided).
- Methods should be better describe to be clear enough.
- The article includes many abbreviations that makes the reading difficult.
- English language punctuation must be improved - the dots instead of commas should be put in the right place in the text.
Author Response
Response to Reviewer 1 Comments
Thank you for your letter dated Mar.4th. We thank the reviewers for the time and effort they have put into reviewing the previous version of the manuscript. Their suggestions have enabled us to improve our work. Based on the instructions provided in your letter, we have made a revised version of the manuscript.
Appended to this letter is our point-by-point response to the comments raised by the reviewers. The comments are reproduced, and our responses are given directly afterward in red color.
We would like also to thank you for allowing us to resubmit a revised copy of the manuscript. And we hope that the revised manuscript could be accepted for publication in Sustainability
Thank you for your important and kind review comments.
Sincerely
Lizhi Xing
Mar. 9th, 2022
Email: [email protected]
Point 1: The title of the article is too long and it is very narrow which would suggest the article is not interesting for international readers.
Response 1: Thank you very much for your valuable suggestions, we have changed the title to:
Measuring the Intermediate Goods’ External Dependency on the Global Value Chain: A Case Study of China
Point 2: This presents the China case mainly and a broader perspective and separate literature review part are needed.
Response 2: Thank the Reviewer for pointing out the shortcomings of this part. We agree that providing a broader perspective and separate literature review part is really necessary. We have rewritten the Introduction and Literature Review sections.
Point 3: The introduction part focuses on politics mainly. This should be avoided to quote political statements in a scientific article.
Response 3: Thank you for your profound advice. We have removed political statements.
Point 4: The scientific problem and the novelty contribution to science should be better presented.
Response 4: Thank you for your valuable comments, we have highlighted the scientific questions and contributions of this paper in the new edition of the manuscript. The relevant content is as follows:
Based on the above viewpoints, we believe that it is necessary to re-examine the dependence of each country's production system on its domestic value chain and foreign value chain and its economic significance from a global perspective. Therefore, the scientific problems to be solved in this paper include two: one is to construct a GIVCN network model reflecting the topological characteristics of the global value chain based on complex network theory and multi-regional input-output (MRIO) data and to restore the industrial sector level of the world's major economies The second is to measure the Degree of Vertical Specialization of the industrial sector (VSD) and the Import Share of Domestic Total Consumption (IMS) based on the concept of trade intermediary attributes. The two indicators proposed in this study enrich the theoretical framework of the research on the sustainability of industrial development This paper takes the “dual circulation” perspective as the starting point, measures the degree of dependence on China's manufacturing industry under the global value chain division of labor system. Finally, this paper puts forward relevant suggestions on achieving a two-way balance between foreign countries and Chinese regions and sustainable development.
Point 5: The topic of the manuscript refers closely to some current issues, e.g. Covid-19 pandemic impact on global trade and production. The current Covid -19 pandemic issue should be better described and extended in the manuscript. The Covid-19 issue is mentioned once in the article and this does even not refer to the scientific research (no reference is provided).
Response 5: Thank you for your valuable comments. Indeed, the context of the COVID-19 pandemic is crucial to the formulation of the scientific question of this paper. Therefore, this article provides further relevant discussions and literature references. As follows:
The world has fallen into the trend of "de-globalization", and the sluggish global value chain has made the environment for countries to participate in the international division of labor even worse. As a result, from the perspective of production, the division of labor in the global value chain led and promoted by multinational corporations has shrunk to a certain extent [1][2]. In particular, the COVID-19 epidemic has highlighted the urgency and importance of maintaining the stability of the supply chain under the impact of the "black swan" event [3][4][5]. In the production of intermediate products, industries that are highly dependent on the global supply chain system will seriously affect economic development once they face a decline in the inventory of intermediate products and a lack of alternative channels. Agriculture and food exports from Australia and New Zealand have been largely caught up in delays in freight services due to the emergence of COVID-19. Achieving resilience in their global value chains is therefore now a key strategy for Australian and New Zealand companies. The key drivers of resilient supply chains are domestic production systems and consumer markets, and industrial exports should consider reducing excessive dependence on other countries [6].
- McKinsey Global Institute. Globalization in transition: The future of trade and value chains [J]. 2019.
- Dollar D. Technological innovation, supply chain trade, and workers in a globalized world [J]. Global Value Chain Development Report, 2019, 1.
- Kapoor K, Bigdeli A Z, Dwivedi Y K, et al. How is COVID-19 altering the manufacturing landscape? A literature review of imminent challenges and management interventions [J]. Annals of Operations Research, 2021: 1-33.
- Deshmukh S G, Haleem A. Framework for manufacturing in post-COVID-19 world order: An Indian perspective [J]. International Journal of Global Business and Competitiveness, 2020, 15(1): 49-60.
- Hald K S, Coslugeanu P. The preliminary supply chain lessons of the COVID-19 disruption—What is the role of digital technologies? [J]. Operations Management Research, 2021: 1-16.
- Gao H, Ren M. Overreliance on China and dynamic balancing in the shift of global value chains in response to global pandemic COVID-19: an Australian and New Zealand perspective [J]. Asian Business & Management, 2020, 19(3): 306-310.
Point 6: Methods should be better described to be clear enough.
Response 6: Thank you very much for your suggestions. This article supplements the original method part in order to make it easier for readers to understand the method framework. The added explanation of trade types on the GVC is as follows:
While international trade has grown dramatically in the last half-century, the nature of trade has been through a dramatic change. One of the most important changes involves the increasing interconnectedness of production processes in a vertical trading chain that stretches across many countries, each of which specializes in particular stages of production. The main feature of vertical specialization is a country imports intermediate goods as inputs from another country, and then its inputs are translated with value-added into outputs that are exported to the third country. This production process includes numerous domestic and foreign trades about intermediate goods, and it ends until final goods arrive at the consumer market.
Trade can be classified into four types according to whether commerce and trade economy happen within the same country or not and whether the exchange of goods and services happens between different sectors or not. Then, classification in detail has also been made based on two dimensions shown in Figure 3: Inter-Industry Trade, Intra-Industry Trade, Industrial Input-Output Trade, and Industrial Self-Consumption Trade.
Figure 1. Four Trade Types in Quadrants
Point 7: The article includes many abbreviations that make the reading difficult.
Response 7: I'm very sorry for the difficulty in reading because of the use of too many abbreviations in the original text. We have removed abbreviations from the original text, except for the most frequently used ones. We have listed the most commonly used abbreviations for your reference:
Import Share of Domestic Total Consumption: IMS
Degree of Vertical Specialization: VSD
Multi-regional Input-output Database: MRIO
Global Value Chain: GVC
Global Industrial Value Chain Network: GIVCN
Social Network Analysis: SNA
Trade Brokerage Property: TBP
Primary (P) sectors, Low Tech (LT) sectors, and High and Medium Tech (HMT)
Point 8: English language punctuation must be improved - the dots instead of commas should be put in the right place in the text.
Response 8: Thank you very much for your valuable comments, we have optimized the language of the full text.
All the changes made to the manuscript are contained in the “Revised Manuscript” and “Revised Manuscript with Change Tracks” files.
Thank you very much!

Reviewer 2 Report
This paper is interesting. However, I would like to make a few critical remarks:
1) In the introduction, the scientific positions explaining the trends in the world economy in the global environment under analysis should be strengthened, we need to see a broader context on a Product Production and Circulation scientific discussion, not only on the Chinese subject.
Please appreciate this aspect as it would be of most interest to the global scientific community.
2) The wording of the scientific problem should also be strengthened and expressed in question (s): RQ1; RQ2. This is the main methodological requirement in the construction of a scientific problem.
3) The conceptualization of the main criteria should also be presented and justified. This would make the instrumentation of the study much clearer.
4) Again, what is the scientific value of this article and the main scientific result?
5) It would be necessary to provide section of the discussion: to justify what is the difference comparing your study and the studies of other researchers in this area.
Modelling and Framework is good.
Author Response
Response to Reviewer 2 Comments
Thank you for your letter dated Mar.4th. We thank the reviewers for the time and effort they have put into reviewing the previous version of the manuscript. Their suggestions have enabled us to improve our work. Based on the instructions provided in your letter, we have made a revised version of the manuscript.
Appended to this letter is our point-by-point response to the comments raised by the reviewers. The comments are reproduced, and our responses are given directly afterward in red color.
We would like also to thank you for allowing us to resubmit a revised copy of the manuscript. And we hope that the revised manuscript could be accepted for publication in Sustainability
Thank you for your important and kind review comments.
Sincerely
Lizhi Xing
Mar. 9th, 2022
Email: [email protected]
Point 1: In the introduction, the scientific positions explaining the trends in the world economy in the global environment under analysis should be strengthened, we need to see a broader context on a Product Production and Circulation scientific discussion, not only on the Chinese subject.
Please appreciate this aspect as it would be of most interest to the global scientific community.
Response 1: Thank the Reviewer for pointing out the shortcomings of this part. We have provided a broader international context in the new edition of the paper, as follows:
In the context of economic globalization, multinational enterprises integrate production resources scattered around the world according to the resource endowments and comparative advantages of countries or regions. By reconfiguring each process in the production process in different countries or regions, an international vertical specialization division of labor system is finally formed [1]. Under the existing international division of labor system, the development of a country or region's production system not only depends on its own final demand and production technology but also on its position in the international division of labor system [2][3]. In the era of globalization, the economies of various countries are interconnected through trade and exchange rates. Therefore, an economic or financial crisis in a major economy can also adversely affect other countries. For example, the occurrence of the subprime mortgage crisis in the United States not only affected the US economy but also caused the market demand in European countries to remain sluggish [4]. Therefore, while actively participating in the global industrial division of labor, countries should also pay attention to the degree of dependence of the industry on the global value chain.
Global value chains are now undergoing severe shocks. The development of digital technology has reduced labor costs and promoted the return of labor-intensive industries from developing countries to developed countries. In addition, the world has fallen into the trend of "de-globalization", and the sluggish global value chain has made the environment for countries to participate in the international division of labor even worse. As a result, from the perspective of production, the division of labor in the global value chain led and promoted by multinational corporations has shrunk to a certain extent [5][6]. In particular, the COVID-19 epidemic has highlighted the urgency and importance of maintaining the stability of the supply chain under the impact of the "black swan" event [7][8][9]. In the production of intermediate products, industries that are highly dependent on the global supply chain system will seriously affect economic development once they face a decline in the inventory of intermediate products and a lack of alternative channels. Agriculture and food exports from Australia and New Zealand have been largely caught up in delays in freight services due to the emergence of COVID-19. Achieving resilience in their global value chains is therefore now a key strategy for Australian and New Zealand companies. The key drivers of resilient supply chains are domestic production systems and consumer markets, and industrial exports should consider reducing excessive dependence on other countries[10].
More and more countries are realizing the importance of industrial chain security. In order to coordinate the safety of the global layout of the industrial chain and the sustainability of the production cycle of intermediate products within the country, the country tends to complete a three-stage production cycle domestically. In other words, the cycle of production is to increase the domestic production cycle of intermediate products. However, whether the transfer of intermediate production to the country can be successful depends to a large extent on the integrity of the current domestic production system and the degree of external dependence. Therefore, analyzing the dependence of the country on the global industrial chain to complete the domestic intermediate product production under the existing industrial layout will help to avoid the risks of global intermediate product trade caused by uncertainty, technology blockade, rule decoupling and other major hidden dangers.
- Martin-Montaner J A, Ríos V O. Vertical specialization and intra-industry trade: The role of factor endowments [J]. Weltwirtschaftliches Archiv, 2002, 138(2): 340-365.
- Meng B, Inomata S. Production networks and spatial economic interdependence: An international input-output analysis of the Asia-Pacific region [M]. Inst. of Developing Economies, Japan External Trade Organization, 2009.
- McWilliam, S.E., Nielsen, B.B. Global value chains and development: Redefining the contours of 21st century capitalism. J Int Bus Stud 51, 1347–1350 (2020). https://doi.org/10.1057/s41267-020-00303-3.
- Curran L, Escaith H, Hallaert J J, et al. The impact of the financial and economic crisis on world trade and trade policy [J]. Intereconomics, 2009, 44(5): 264-293.
- McKinsey Global Institute. Globalization in transition: The future of trade and value chains [J]. 2019.
- Dollar D. Technological innovation, supply chain trade, and workers in a globalized world [J]. Global Value Chain Development Report, 2019, 1.
- Kapoor K, Bigdeli A Z, Dwivedi Y K, et al. How is COVID-19 altering the manufacturing landscape? A literature review of imminent challenges and management interventions [J]. Annals of Operations Research, 2021: 1-33.
- Deshmukh S G, Haleem A. Framework for manufacturing in post-COVID-19 world order: An Indian perspective [J]. International Journal of Global Business and Competitiveness, 2020, 15(1): 49-60.
- Hald K S, Coslugeanu P. The preliminary supply chain lessons of the COVID-19 disruption—What is the role of digital technologies? [J]. Operations Management Research, 2021: 1-16.
- Gao H, Ren M. Overreliance on China and dynamic balancing in the shift of global value chains in response to global pandemic COVID-19: an Australian and New Zealand perspective [J]. Asian Business & Management, 2020, 19(3): 306-310.
Point 2: The wording of the scientific problem should also be strengthened and expressed in question (s): RQ1; RQ2. This is the main methodological requirement in the construction of a scientific problem.
Response 2: Thank you for your valuable comments, we have highlighted the scientific questions of this paper in the new edition of the manuscript. The relevant content is as follows:
Based on the above viewpoints, we believe that it is necessary to re-examine the dependence of each country's production system on its domestic value chain and foreign value chain and its economic significance from a global perspective. Therefore, the scientific problems to be solved in this paper include two: one is to construct a GIVCN network model reflecting the topological characteristics of the global value chain based on complex network theory and multi-regional input-output (MRIO) data and to restore the industrial sector level of the world's major economies The second is to measure the Degree of Vertical Specialization of the industrial sector (VSD) and the Import Share of Domestic Total Consumption (IMS) based on the concept of trade intermediary attributes. The two indicators proposed in this study enrich the theoretical framework of the research on the sustainability of industrial development This paper takes the “dual circulation” perspective as the starting point, measures the degree of dependence on China's manufacturing industry under the global value chain division of labor system. Finally, this paper puts forward relevant suggestions on achieving a two-way balance between foreign countries and Chinese regions and sustainable development.
Point 3: The conceptualization of the main criteria should also be presented and justified. This would make the instrumentation of the study much clearer.
Response 3: Thank you very much for your suggestions. This article supplements the original method part in order to make it easier for readers to understand the method framework. The added explanation of trade types on the GVC is as follows:
While international trade has grown dramatically in the last half-century, the nature of trade has been through a dramatic change. One of the most important changes involves the increasing interconnectedness of production processes in a vertical trading chain that stretches across many countries, each of which specializes in particular stages of production. The main feature of vertical specialization is a country imports intermediate goods as inputs from another country, and then its inputs are translated with value-added into outputs that are exported to the third country. This production process includes numerous domestic and foreign trades about intermediate goods, and it ends until final goods arrive at the consumer market.
Trade can be classified into four types according to whether commerce and trade economy happen within the same country or not and whether the exchange of goods and services happens between different sectors or not. Then, classification in detail has also been made based on two dimensions shown in Figure 1: Inter-Industry Trade, Intra-Industry Trade, Industrial Input-Output Trade, and Industrial Self-Consumption Trade.
Figure 1. Four Trade Types in Quadrants
Point 4: Again, what is the scientific value of this article and the main scientific result?
Response 4: Thank you for your valuable comments, we have highlighted the scientific value and important contributions of this paper in a new version of the paper. The relevant content is as follows
Based on the above viewpoints, we believe that it is necessary to re-examine the dependence of each country's production system on its domestic value chain and foreign value chain and its economic significance from a global perspective. Therefore, the scientific problems to be solved in this paper include two: one is to construct a GIVCN network model reflecting the topological characteristics of the global value chain based on complex network theory and multi-regional input-output (MRIO) data and to restore the industrial sector level of the world's major economies The second is to measure the Degree of Vertical Specialization of the industrial sector (VSD) and the Import Share of Domestic Total Consumption (IMS) based on the concept of trade intermediary attributes. The two indicators proposed in this study enrich the theoretical framework of the research on the sustainability of industrial development This paper takes the “dual circulation” perspective as the starting point, measures the degree of dependence on China's manufacturing industry under the global value chain division of labor system. Finally, this paper puts forward relevant suggestions on achieving a two-way balance between foreign countries and Chinese regions and sustainable development.
The existing literature on the division of labor in the global value chain constructs indicators from the perspective of trade added value, which reflects the amount of domestic value contained in export products. The method proposed in this paper is based on social network analysis. The intermediary attribute of the intermediary is an in-depth study of the cross-border production model in the trade of intermediate goods, which complements the analysis angle based on value-added. Although the above literature studies globalization and analyzes the external dependence of value-added trade from a multi-dimensional perspective, there are few studies to quantify and analyze the trade pattern of internal and external circular production of national intermediate goods and the degree of dependence of different industries on the global value chain. In the case of failing to accurately grasp the basic laws of directional selection inside and outside the economic cycle, local governments are easily caught in the dilemma of the blind layout. And this paper can provide a certain theoretical basis for the country to rearrange the industrial chain.
Compared with the existing literature, the innovation of this paper is mainly reflected in the following aspects: (1) This study uses a multi-regional input-output model to provide a new measurement framework for the classification of trade patterns based on intermediary attributes. This measurement framework enhances the study of international trade from an economic physics perspective. (2) Based on the characteristics of the global input-output model, this paper defines Degree of Vertical Specialization (VSD), Import Share of Domestic Total Consumption (IMS) indicators. (3) In response to the "dual circulation" policy proposed by China, the external dependence measurement of the global intermediate product production cycle is extended, and the time evolution law of the domestic market participating in the internal and external circulation is explored. The technical framework provided in this paper is conducive to accurately capturing the key industries under the "dual circulation" strategy and puts forward relevant suggestions on reducing the losses caused by the shift of the industrial chain by analyzing the characteristics of the industries. The arrangement of this paper is as follows: Section 3 discusses the modeling process of the global industrial value chain network, Section 4 defines the intermediary attributes of the industrial sector and gives a method for measuring dependence, Section 5 conducts empirical research and discusses, and Section 6 presents the promotion of china's policy recommendations for the implementation of the "dual circulation" strategy.
Point 5: It would be necessary to provide a section of the discussion: to justify what is the difference comparing your study and the studies of other researchers in this area.
Response 5: Thank you very much for pointing out the shortcomings of this article. It is necessary to increase the discussion in the text, so we add a discussion of the empirical results and methodology of this paper in the text. The content of this part has been updated in the new version.
All the changes made to the manuscript are contained in the “Revised Manuscript” and “Revised Manuscript with Change Tracks” files.
Thank you very much!

Round 2
Reviewer 1 Report
Dear author/s,
Thank you very much for your detailed answers to all my suggestions and revised version of the manuscript!
Good luck!
reviewer